# Identification of High-Risk Groups in Urinalysis: Lessons from the Longitudinal Analysis of Annual Check-Ups

**DOI:** 10.3390/healthcare10091704

**Published:** 2022-09-06

**Authors:** Keiichi Matsuzaki, Tomohiro Ohigashi, Takashi Sozu, Mami Ishida, Daisuke Kobayashi, Hitoshi Suzuki, Yusuke Suzuki, Takashi Kawamura

**Affiliations:** 1Agency for Health, Safety and Environment, Kyoto University, Kyoto 606-8501, Japan; 2Department of Biostatistics, Tsukuba Clinical Research & Development Organization, University of Tsukuba, Ibaraki 305-8576, Japan; 3Department of Information and Computer Technology, Faculty of Engineering, Tokyo University of Science, Tokyo 125-8585, Japan; 4Department of Preventive Services, School of Public Health, Graduate School of Medicine, Kyoto University, Kyoto 606-8303, Japan; 5Department of Nephrology, Juntendo University Faculty of Medicine, Tokyo 113-8421, Japan

**Keywords:** urinalysis, mass screening, annual health check-up

## Abstract

Background: For effective screening in urinalysis, information on high-risk groups is needed; however, there is a lack of evidence in young adults in particular. The aim of this study was to provide information on urinalysis in young adults and to identify high-risk groups of urinalyses using multi-year data obtained from annual large-scale check-ups. Method: We used annual health check-up data collected from 2011 to 2016 at Kyoto University in Japan. Eligible participants were those aged 18–39 years who underwent annual health check-ups for four consecutive years between 2011 and 2016. We conducted descriptive analyses and calculated the risk ratios (RRs) for urinary abnormalities in the fourth year of urinalysis. Results: In total, 13,640 participants (10,877 men, 79.7%) met the eligibility criteria. The mean prevalence rates of proteinuria, haematuria and glucosuria were 1.61% (men: 1.63%; women: 1.53%), 1.48% (men: 0.53%; women: 5.22%) and 0.46% (men: 0.52%; women: 0.25%), respectively. Participants with urinary abnormalities at least once in the initial 3 years had a higher risk of urinary abnormalities in the fourth year than participants with no abnormal findings in the initial 3 years; the risk ratios (RRs) of proteinuria, haematuria and glucosuria were 3.5 (95% confidence interval (CI) = 3.2–3.7), 12.2 (95% CI = 11.7–12.7) and 42.6 (95% CI = 37.7–48.1), respectively. The RRs of all urinary abnormalities in the fourth year increased as the frequency of urinary abnormalities over the preceding 3 years increased. In haematuria, differences of the RR were observed between men and women. Conclusion: We clarified the prevalence of urinary abnormalities in young adults and high-risk groups of urinary abnormalities. Our findings support the need for multi-year annual urinalysis.

## 1. Introduction

Urinalysis, which can be performed easily and at low cost, is essential for the early detection of renal diseases with few subjective symptoms. Since 1972, a system of annual urinalysis for workers that was introduced by the Ministry of Health, Labour and Welfare has been in effect in Japan [1]. Furthermore, since 1973, a screening programme for haematuria and proteinuria for school children and young adolescents that was introduced by the Ministry of Education, Science and Culture has been in effect in Japan. In 1992, due to the increase in the prevalence of childhood obesity, a urinary glucose test was added to the annual check-up of children to detect early-stage diabetes mellitus [2]. Several studies that used general screening cohorts have reported relationships between proteinuria and mortality [3,4], between haematuria and mortality [5], and between glucosuria and mortality [6].

However, urinalysis is known as an inaccurate test because the findings of urinalysis are often affected by proteinuria, cystitis, vitamin C intake and physical conditions such as upright posture. Thus, unfavourable results in cost–performance analyses of annual urinalysis screening have been reported in several studies [7,8,9]. Boulware et al. reported that annual urinalysis screening is not effective unless performed among high-risk persons (such as the elderly or patients with hypertension) or conducted frequently at intervals of 10 years [7], and a systematic review suggested that screening for CKD in the general population without risk stratification is unlikely to be cost-effective [10].

For effective screening, risk stratification of urinalysis is needed. Several studies have reported urinalyses that included participants in children [11,12,13] and those over 40 years of age [4,5,6]; however, there is limited information on urinalysis in young adults. The aim of this study was to provide information on urinary abnormalities in young adults and to identify groups with a high incidence of urinary abnormalities using a large-scale database of multi-year annual health check-ups.

## 2. Materials and Methods

### 2.1. Annual Health Check-Ups

In Japan, the School Health and Safety Act requires schools to provide annual health check-ups for students, and the Industrial Safety and Health Act requires employers to provide annual health check-ups for employees. Kyoto University has about 23,000 students and 10,000 employees, and the health service at the university conducts annual health check-ups for students and employees in April and September, respectively. The annual health check-up for the students and employees of Kyoto University consists of anthropometry, laboratory tests including urine and blood examinations, and a self-administered questionnaire survey of individual disease history, and lifestyle and family health history. Height and body weight were evaluated simultaneously using a scale that features an automatic measurement device (KS-610NF, Kansai Seiki Co., Ltd., Kyoto, Japan). Body mass index was calculated as body weight (in kilograms) divided by the square of the height (in metres). Systolic blood pressure and diastolic blood pressure were measured using an automatic sphygmomanometer (OMRON HEALTHCARE Co., Ltd., Kyoto, Japan). Dipstick urinalysis was performed using Uropisu S (Minaris Medical Co., Ltd., Tokyo, Japan). Using a colour scale, proteinuria, haematuria and glucosuria were classified as negative, trace, 1+, 2+, 3+ or 4+. If the participants were in the menstrual period, urinalysis was performed later. Data on disease history, smoking status, alcohol consumption and physical activity were collected using a self-reported questionnaire.

### 2.2. Study Participants

We extracted the annual health check-up data of students and employees aged 18 to 39 who underwent four consecutive check-ups between 2011 and 2016 from the database. Participants who had a history of urinary stones, kidney disease and diabetes mellitus or impaired glucose tolerance at baseline were excluded.

### 2.3. Statistical Analysis

Continuous variables were expressed as mean ± standard deviation (SD). Urinary abnormality was defined as Class 1+ or higher for each urinalysis item. Mean prevalence was defined as the average incidence rate in each year. We calculated the proportion of urinary abnormalities in the fourth year for participants with a urinary abnormality at least once in the initial 3 years (p1), those with no urinary abnormality in the initial 3 years (p2), those with urinary abnormalities twice or more in the initial 3 years (p3), those with urinary abnormalities once or no time in the initial 3 years (p4), those with urinary abnormalities thrice in the initial 3 years (p5) and those with urinary abnormalities twice or less in the initial 3 years (p6). Thereafter, we calculated the risk ratios (RRs: p1/p2, p3/p4, p5/p6), their respective 95% confidence intervals (CIs) and the *p*-values of the Fisher’s exact test. The RRs of proteinuria, haematuria and glucosuria were calculated for all participants and for each sex. All CIs were calculated at the 95% level. All analyses were performed using SAS (version 9.4; SAS Institute Inc., Cary, NC, USA).

### 2.4. Ethical Issues

In accordance with the national ethical guidelines of Japan, individual informed consent was waived for the use of anonymised clinical data. Instead, an opt-out approach was used in this study. All personal identifiers were removed before the data were provided to the researchers. The Ethics Committee of Kyoto University Graduate School and the Faculty of Medicine approved the study protocol (registration number: R1041).

## 3. Results

In total, 13,640 participants met the eligibility criteria. Table 1 summarises the demo-graphic, anthropometric, medical history and lifestyle characteristics of the study participants; 10,877 participants (79.7%) were men, and the mean (±SD) age of the participants was 21.7 (±5.3) years. 

Table 2 shows the mean prevalence of urinary abnormalities and the prevalence of abnormalities in two or more consecutive years for proteinuria, haematuria and glucosuria. The mean prevalence of proteinuria, haematuria and glucosuria were 1.61% (men: 1.63%; women: 1.53%), 1.48% (men: 0.53%; women: 5.22%) and 0.46% (men: 0.52%; women: 0.25%), respectively. The prevalence of abnormalities in two or more consecutive years was 0.26% for proteinuria (men: 0.25%; women: 0.33%), 0.66% for haematuria (men: 0.36%; women: 1.85%) and 0.15% for glucosuria (men: 0.17%; women: 0.07%).

Table 3 shows the associations between the presence of urinary abnormalities in the fourth year and their frequency in the initial 3 years. The RRs of all urinary abnormalities in the fourth year increased as the frequency of the urinary abnormalities over the preceding 3 years increased (RR for proteinuria: once or more: 3.5, twice or more: 9.9; RR for haematuria: once or more: 12.2, twice or more: 23.3, three times or more: 47.1; RR for glucosuria: once or more: 42.6, twice or more: 112.5, three times or more: 223.5). The RRs of proteinuria and glucosuria in men and women were comparable. In contrast, differences in the RR for haematuria were observed between men and women (once or more: 73.2 in men and 2.4 in women; twice or more: 113.6 in men and 5.2 in women; three times or more: 171.3 in men and 14.7 in women).

The supplementary table shows the details of number of participants according to the patterns of urinary abnormalities. The prevalence of urinary abnormalities in the fourth year according to the frequency of urinary abnormalities in the initial 3 years is described below. Proteinuria was found in the fourth year in 19.0% of participants who had proteinuria twice in the initial 3 years, in 4.7% of participants who had proteinuria only once in the initial 3 years and in 1.5% of participants who had no proteinuria in the initial 3 years (Appendix A). One participant who had presented with proteinuria thrice in the initial 3 years showed the onset of kidney disease in the second year. This participant had both haematuria and proteinuria in the first year; these findings disappeared in the third year and fourth year, respectively. Haematuria was found in the fourth year in 64.7% of participants who had haematuria thrice in the initial 3 years, in 21.1% of participants who had haematuria twice in the initial 3 years, in 8.9% of participants who had haematuria only once in the initial 3 years and in 1.0% of participants who had no haematuria in the initial 3 years (Appendix A). Glucosuria was found in the fourth year in all of participants who had glucosuria thrice in the initial 3 years, in 33.3% of participants who had glucosuria twice in the initial 3 years, in 8.6% of participants who had glucosuria only once in the initial 3 years and in 0.3% of participants who had no glucosuria in the initial 3 years (Appendix A). One participant who had demonstrated glucosuria thrice in the initial 3 years showed the onset of diabetes mellitus in third year. This participant was started on treatment for diabetes mellitus during third year, and the glucosuria was absent in the fourth year.

## 4. Discussion

Our study clarified the prevalence of urinary abnormalities and identified the high-risk groups of urinary abnormalities using multi-year annual check-ups. This study also revealed that the prevalence of urinary abnormalities increases as the frequency of urinary abnormalities increases over time. These findings may lead to more efficient urinalysis during annual health check-ups by detecting the high-risk groups of urinary abnormalities. To the best of our knowledge, this is the first study to describe the results of multi-year annual check-up urinalysis in young adults.

Since most primary forms of chronic glomerulonephritis first manifest as asymptomatic proteinuria and/or haematuria [14,15], urinalysis is considered one of the best methods for the early detection of glomerulonephritis [11]. Moreover, large-scale cohort studies [3,16] and meta-analyses [17] of the general population revealed that proteinuria is a significant risk factor of end-stage kidney disease (ESKD), cardiovascular mortality and all-cause mortality. However, the need for urinalysis is still controversial when considered in terms of cost-effectiveness [10]. Boulware et al. reported that urinary screening of US adults is not cost-effective unless selectivity is directed towards high-risk groups [7]. In this study, we found that proteinuria occurring at least once in the initial 3 years is associated with a high risk of proteinuria in the fourth year. We also found that the RR of proteinuria increased as the frequency of proteinuria increased, and this trend observed with proteinuria was found to be similar to that observed with haematuria. Our study results clearly identified groups at high risk of proteinuria and/or haematuria, and can inform the establishment of a cost-effective and efficient urinalysis system.

The annual incidence rate of proteinuria has been reported in several studies. Nagai et al. reported the annual incidences of proteinuria and persistent proteinuria, defined as proteinuria that occurs over two consecutive years, in subjects over 40 years of age [18]. They stated that the annual incidence of proteinuria was 1.31% in men and 0.68% in women. Furthermore, they reported that the annual incidence of persistent proteinuria was 0.33% in men and 0.14% in women. In this study of young adults, we found that the mean prevalence rate of proteinuria was 1.63% in men and 1.53% in women, and that the incidence of persistent proteinuria was 0.25% in men and 0.33% in women. Although the participants in our study had a lower incidence of risk factors for persistent proteinuria (e.g., hypertension, diabetes or reduced renal function), almost similar results were obtained. At present, the rationale for this result remains unclear. Future prospective cohort studies are needed to investigate the risk factors for persistent proteinuria in young adults.

Persistent asymptomatic haematuria is a known risk factor of ESKD in adolescents and young adults [19]. Recently, Iseki et al. reported that dipstick haematuria was associated with high all-cause mortality in men [5]. Additionally, persistent dipstick haematuria was reported to be a significant risk factor of all-cause mortality in men [20]. The results of our study suggest that the incidence of haematuria at least once in the initial 3 years was associated with a high risk of haematuria in the fourth year. Interestingly, the mean prevalence rate of haematuria was found to be higher in women than in men, but the RR of haematuria was found to be higher in men than in women. We expect that our study findings will promote multi-year screening for haematuria.

Glucosuria is the result of glycaemic excursions above the renal glucose threshold. Since urine glucose level is not a sensitive measure for diabetes mellitus screening, the use of glucosuria in the screening programme is limited. Glycated haemoglobin (HbA1c) level is the standard marker for diabetes mellitus screening; however, it is impractical for mass screening due to the associated high cost of measurement. Therefore, the World Health Organization suggested that the urine glucose level should be one of the variables measured during diabetes mellitus screening in low-resource settings [21]. Similar to participants with proteinuria and participants with haematuria, participants with glucosuria that occurred at least once in the initial 3 years were found to be at high risk for glucosuria in the fourth year. This result clarified the high-risk group of glucosuria, which may be useful for identifying target groups for diabetes mellitus screening in low-resource settings.

This study has several limitations. First, since the items to be assessed during the annual check-up are few, we could not evaluate disease incidence. To overcome this drawback, we are considering conducting a prospective cohort study with a new questionnaire at the university. Second, since our study participants included young adults, urinary abnormalities were rare. Thus, individual urinary abnormalities may have been overestimated. Third, in addition to increased blood sugar levels, glucosuria can be caused by temporary glucose excretion via urine such as is observed following treatment with a sodium-glucose co-transporter 2 inhibitor. Unfortunately, we did not collect data on such treatments in our study. It is necessary to conduct further studies to determine the diagnostic accuracy of diabetes mellitus screening based on measurements of urine glucose levels.

## 5. Conclusions

We clarified the prevalence of urinary abnormalities in young adults and identified groups at high risk of urinary abnormalities using data from multi-year annual check-up urinalysis. Our study findings support the need for multi-year annual check-up urinalysis, which can help clinicians and health policymakers make more effective decisions to the benefit of the participants they serve.

## Figures and Tables

**Table 1 healthcare-10-01704-t001:** Demographic, anthropometric, medical history and lifestyle characteristics of participants in this study.

	Total (*n* = 13,640)	Missing
Age, years (SD)	21.7 (5.3)	0
Male sex (%)	10,877 (79.7)	0
Height, cm (SD)	169.2 (7.6)	0
Weight, kg (SD)	60.9 (10.2)	2
BMI	21.2 (2.9)	2
Systolic blood pressure, mmHg (SD)	124.9 (14.8)	0
Diastolic blood pressure, mmHg (SD)	72.3 (10.2)	0
Prevalence of proteinuria in the first year (%)	200 (1.5)	0
Prevalence of haematuria in the first year (%)	184 (1.4)	0
Prevalence of glucosuria in the first year (%)	46 (0.3)	0
History of hypertension		0
None (%)	13,599 (99.7)	
In treatment, under observation or in the past (%)	41 (0.3)	
History of dyslipidaemia		0
None (%)	13,609 (99.8)	
In treatment, under observation or in the past (%)	31 (0.2)	
Smoking history		17
None (%)	13,277 (97.3)	
Smoking (%)	345 (2.5)	
Alcohol consumption		16
Seldom or never (%)	13,144 (96.4)	
Every day or often (%)	480 (3.5)	
Frequency of exercise		28
Every day (%)	971 (7.1)	
Sometimes (%)	6167 (45.2)	
seldom (%)	6474 (47.5)	

Abbreviations: BMI: body mass index, SD: standard deviation.

**Table 2 healthcare-10-01704-t002:** The mean prevalence and the prevalence of abnormalities in two or more consecutive years of proteinuria, haematuria and glucosuria.

	All (*n* = 13,640)	Men (*n* = 10,877)	Women (*n* = 2763)
Mean prevalence			
Proteinuria (%)	1.61	1.63	1.53
Haematuria (%)	1.48	0.53	5.22
Glucosuria (%)	0.46	0.52	0.25
Prevalence of abnormalities in two or more consecutive years			
Proteinuria (%)	0.26	0.25	0.33
Haematuria (%)	0.66	0.36	1.85
Glucosuria (%)	0.15	0.17	0.07

**Table 3 healthcare-10-01704-t003:** Associations between prevalence of urinary abnormalities in the fourth year and their frequency in the initial 3 years.

	All (*n* = 13,640)			Men (*n* = 10,877)			Women (*n* = 2763)		
Frequency of Abnormalities in the Initial 3 Years	RR	95% CI	*p*-Value	RR	95% CI	*p*-Value	RR	95% CI	*p*-Value
Proteinuria									
Once or more (versus none)	3.5	3.2 to 3.7	<0.001	3.3	3.0 to 3.6	<0.001	4.4	3.2 to 6.1	0.002
Twice or more (versus once and none)	9.9	6.5 to 15.0	<0.001	6.7	2.8 to 16.1	0.036	19.7	9.3 to 41.6	0.005
Three times (versus twice, once and none)	NA	-	-	NA	-	-	NA	-	-
Haematuria									
Once or more (versus none)	12.2	11.7 to 12.7	<0.001	73.2	62.1 to 86.2	<0.001	2.4	2.3 to 2.6	<0.001
Twice or more (versus once and none)	23.3	21.8 to 24.8	<0.001	113.6	96.4 to 134	<0.001	5.2	4.6 to 5.9	<0.001
Three times (versus twice, once and none)	47.1	43.8 to 50.7	<0.001	171.3	144.6 to 202.9	<0.001	14.7	13.2 to 16.4	<0.001
Glucosuria									
Once or more (versus none)	42.6	37.7 to 48.1	<0.001	40.8	35.5 to 46.9	<0.001	49.1	15.7 to 152.9	0.001
Twice or more (versus once and none)	112.5	97.6 to 129.8	<0.001	109.4	93.6 to 127.9	<0.001	115	24.4 to 542.3	0.010
Three times (versus twice, once and none)	223.5	216.5 to 230.8	<0.001	209.1	201.4 to 217.1	<0.001	NA	-	*p-*value

*p*-values were calculated by the Fisher’s exact test. Abbreviations: RR: risk ratio; CI: confidence interval; NA: not applicable.

## Data Availability

The data underlying this article cannot be shared publicly because of the privacy of individuals that participated in the study. The data will be shared on reasonable request to the corresponding author.

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
