# Peer review of "Identification of High-Risk Groups in Urinalysis: Lessons from the Longitudinal Analysis of Annual Check-Ups"

_healthcare, 2022, doi:10.3390/healthcare10091704_

Round 1

Reviewer 1 Report

Major issues:

The group differences are discussed based on the risk-ratios (RRs) being different across the groups, but a related p-value is not shared. It is highly recommended that RRs are shared with CI and related p-values. The authors share CI but no p-values. The authors could also share causal reasons for the prevalence of the conditions over years for a patient or two as example in main text or supplementary, this would be good to share for the scientific community and in addition would show that the analyses/conclusions are backed by causality.

Minor issues:

The table 1 summarizes the medical history and lifestyle characteristics of patients but seems to miss information such as diabetes which is related to glucosuria (one of the conditions shared in the study), please include this information. Authors could provide reasons as to why subjects were selected only in a particular year range 2011-2016 and why not from recent years. Inclusion criteria can be clarified better.

Reviewer 2 Report

This paper aimed to clarify the prevalence of urinary abnormalities in young adults using data coming from multi-year annual check-up urinalysis. The topic is interesting and the results too. The study design is clear and well structured.

My only comment is that the paper needs an extensive English revision, maybe authors should ask an English native speaker to help them with this issue.

Reviewer 3 Report

This is a very interesting study on the significance of urinalysis for detecting urinary anomalies. This is the first study to describe the result of multi-year annual check-up urinalysis in young adults. 
